# Organic Matter Preservation in Ancient Soils of Earth and Mars

**DOI:** 10.3390/life10070113

**Published:** 2020-07-16

**Authors:** Adrian P. Broz

**Affiliations:** Department of Earth Sciences, University of Oregon, Eugene, OR 97405, USA; abroz@uoregon.edu

**Keywords:** astropedology, biosignature detection, astrobiology, martian paleosols, precambrian paleosols, aqueous alteration

## Abstract

The emerging field of astropedology is the study of ancient soils on Earth and other planetary bodies. Examination of the complex factors that control the preservation of organic matter and other biosignatures in ancient soils is a high priority for current and future missions to Mars. Though previously defined by biological activity, an updated definition of soil as planetary surfaces altered in place by biological, chemical or physical processes was adopted in 2017 by the Soil Science Society of America in response to mounting evidence of pedogenic-like features on Mars. Ancient (4.1–3.7 billion year old [Byr]) phyllosilicate-rich surface environments on Mars show evidence of sustained subaerial weathering of sediments with liquid water at circumneutral pH, which is a soil-forming process. The accumulation of buried, fossilized soils, or paleosols, has been widely observed on Earth, and recent investigations suggest paleosol-like features may be widespread across the surface of Mars. However, the complex array of preservation and degradation factors controlling the fate of biosignatures in paleosols remains unexplored. This paper identifies the dominant factors contributing to the preservation and degradation of organic carbon in paleosols through the geological record on Earth, and offers suggestions for prioritizing locations for in situ biosignature detection and Mars Sample Return across a diverse array of potential paleosols and paleoenvironments of early Mars. A compilation of previously published data and original research spanning a diverse suite of paleosols from the Pleistocene (1 Myr) to the Archean (3.7 Byr) show that redox state is the predominant control for the organic matter content of paleosols. Most notably, the chemically reduced surface horizons (layers) of Archean (2.3 Byr) paleosols have organic matter concentrations ranging from 0.014–0.25%. However, clay mineralogy, amorphous phase abundance, diagenetic alteration and sulfur content are all significant factors that influence the preservation of organic carbon. The surface layers of paleosols that formed under chemically reducing conditions with high amounts of iron/magnesium smectites and amorphous colloids should be considered high priority locations for biosignature investigation within subaerial paleoenvironments on Mars.

## 1. Introduction

Paleosols are ancient, fossil soils, created by removal from their soil-forming factors, either because of changes in those factors or by burial. Modes of burial range from the deposition of volcanic ash and lava flows to rapid sedimentation from flooding or landslides. Paleosols are a geological record on Earth of the atmospheric composition, climate, topography and organisms present before soil burial [1]. Importantly, some of the oldest fossils in terrestrial environments are found in paleosols [2,3]. Soils and life have coevolved with one another over geological time scales, because organisms alter the structure, mineralogy, function and morphology of the soil. Modern soils are saturated with a diverse array of microscopic life, often averaging 10^10–^10^11^ bacterial cells and 10^3^ and 10^4^ species per gram of soil [4]. Life leaves trace or body fossils in paleosols and other biosignatures like isotopically light carbon [5] and complex organic matter [2] as evidence of their ancient relationship with soils, which may have extended well back into the Archean [6,7,8,9]. The intrinsic value of paleosols is now realized across the disciplines of soil and planetary sciences; Paleosols on Mars were recently named a high priority location for in situ biosignature investigation [10,11] and Mars Sample Return [12,13]. 

Recently, there has been increasing consideration of the abiotic pathways of pedogenesis (soil creation) on planets other than Earth. These include chemical alteration through leaching of layered sedimentary rocks in subaerial settings on Mars [14,15,16] and a variety of photochemical reactions that create soil minerals [17,18]. If life does not define soil, then the key questions now are when and how did life appear in soils, and what evidence for life can be preserved in paleosols? Some paleosols such as Histosols and gleyed Entisols preserve organic matter beautifully, e.g., with microscopic details of plant structure [19], but oxidized paleosols contain much less organic carbon than comparable modern soils [1]. 

The purpose of this paper is not to determine the nature and properties of organic matter in paleosols of Earth, because one cannot determine the nature of the inputs (e.g., type and amount) of organic materials from the study of paleosol organic matter alone. Instead, the focus here is on the “results” of life on land, which includes the emplacement and storage of highly recalcitrant (resistant to decay) organic matter in paleosols for millions or even billions of years. More specifically, this paper is a first step towards prioritizing and selecting locations within potential paleosols for in situ biosignature investigation (with *Curiosity* Rover, Mars 2020 *Perseverance* Rover; ExoMars 2020) and Mars Sample Return. 

Whether molecular in size or larger, biosignatures are at the mercy of numerous preservation and degradation mechanisms, most often regulated by redox chemistry, the surfaces of minerals, diagenesis, and depth in the soil profile. This review is a first step towards identifying the factors responsible for organic matter preservation and degradation in Earth’s paleosol record from the Pleistocene to the Archean; it seeks to provide insight into similar phenomena which may influence organic matter storage in Noachean (4.1–3.7 Ga) subaerial paleoenvironments on Mars. 

### 1.1. Paleosols on Mars?

There is mounting evidence that paleosols may be preserved in the geological record on Mars [14,20,21,22,23,24]. One problem inhibiting discussion of paleosols on Mars is that they could not be true soils because a biological component has not yet been proven. However, in 2017, the Soil Science Society of America updated their definition of soil to include abiotic soils on planetary bodies: “The layer(s) of generally loose mineral and/or organic material that are affected by physical, chemical, and/or biological processes at or near the planetary surface and usually hold liquids, gases, and biota and support plants” [25]. This updated definition reflects a modern understanding of abiotic pedogenic pathways, which include subaerial low temperature water–rock interactions (hydrolysis), physicochemical weathering phenomena, and the photochemical creation of soil minerals [17], all of which have been documented on Mars [14,26]. Appendix A discusses Mars-relevant pedogenic features and scales of observation (i.e., in situ to orbital).

The surface of Mars is cold, dry and barren today, but there is extensive evidence that liquid water was stable on the surface of the planet in the geologic past. Geomorphic evidence of liquid water includes deltas, valley networks, and dendritic channels [27] which compliment in situ and orbital detections of phyllosilicates and other hydrated minerals and indicate that surface weathering occurred under an early Mars climate. The conditions for surface weathering were better in the past than on Mars today [14,28]. Several climate models favor a wet but cold surface on early Mars [29], on which the volumetric and temporal occurrence of liquid water markedly decreased [30]. Thus, during most of Martian history, only microscopic liquid water (e.g., microscopic liquid brines) may have been present [31,32], though chemical alteration involving perchlorate brines and microscopic liquid films may have been common [33,34]. Importantly, ultraviolet and ionizing radiation can degrade biosignatures at the surface [35] and subsurface [36], respectively, and surface oxidizing salts including perchlorate can encourage complete combustion of organic carbon during thermal and evolved gas analysis [37,38], which is currently the only method employed for the detection of organic carbon on Mars. 

Both in situ [39,40] and orbital remote sensing [22] techniques have been employed for the investigation of sedimentary rocks and potential paleosols on Mars. In situ examinations focus on geochemical interrogations of samples (e.g., Mars Science Laboratory (MSL) instrument suite onboard *Curiosity* rover), while remote sensing techniques [41] examine mineralogical reflectance and absorbance spectroscopy of the Martian surface, primarily in the visible/near infrared (VNIR) range [15,20]. Hawaiian surface weathering studies and speculation about the Martian climate led Ming et al. (1988) to predict the presence of kaolinite on Mars, which has now been confirmed from orbit [22] and from in situ x-ray diffraction [42]. Chemical analysis and imaging of a late Noachean (3.7 Ga) potential paleosol in the Sheepbed mudstone at Yellowknife Bay, Gale Crater reveals phosphorus depletion, vesicular structure, potential periglacial sand wedges, nodularized instead of crystalline sulfate grouped into a gypsic (By horizon), and ptygmatic folding of a deep sulfate dike, which are compatible with soil that formed under a hyperarid, frigid climate which was later buried by approximately 5 km of overburden [26]. However, the interpretation of this unit as a potential paleosol has been disputed by MSL sedimentologists, who argue the Sheepbed mudstone is a lacustrine sedimentary deposit with subsequent diagenetic alteration, which is consistent with the observed chemical and physical features of the unit.

Sequences of candidate paleosols can also be observed from orbit with visible/near infrared spectroscopy (VNIR). The Compact Reconnaissance Spectrometer for Mars (CRISM, [41]) onboard the Mars Reconnaissance Orbiter is a high-resolution (~18 m/pixel) VNIR (0.35–3.92 μm) imaging spectrometer which has detected what appear to be dioctahedral phyllosilicates in thousands of locations across the surface of the planet [14]. Noachean (4.1–3.7 Ga) terrains at Mawrth Vallis have vertical sequences of dioctahedral phyllosilicate clays which resemble sequences of paleosols on Earth [15]. Paleosols are generally recognized based on clay mineralogy and the stratigraphic distribution of clay minerals, which are sensitive to VNIR spectroscopy [20,43]. Fortunately, many pedogenic minerals are recognized with VNIR spectroscopy due to the strong absorption of the overtones of SO_4_^2−^, CO_3_^2−^ and OH^−^, and combinations of H_2_O and CO_2_ [44]. Common soil and paleosol minerals include hydrated dioctahedral phyllosilicate clays (montmorillonite, illite, nontronite) and amorphous or poorly crystalline phases (e.g., imogolite, ferrihydrite and allophane). Dioctahedral Fe/Mg smectite clays including montmorillonite have characteristic absorption features at 1.9 and 2.3 μm [45,46], and similarities between terrestrial paleosols and subaerial clay deposits on Mars are known from these absorbance features (Figure 1).

Orbital remote sensing of the Martian surface has shown stratigraphic distribution of what appear to be pedogenic minerals similar to those observed in terrestrial sequences of paleosols or leaching profiles (Figure 2) [20,24,47,48]. Potential paleosol sequences in the Arabia Terra region have been documented, and feature layered dioctahedral clay units up to 200 m in thickness [14,23,48], now known as the Mawrth Vallis Group [46]. Some areas show spectral evidence of sediments that have been so intensely weathered that they resemble lateritic soils because of inferred abundances of oxides and Al-rich clays [49]. Postdepositional leaching profiles rarely exceed 100 m in thickness, but the upper limit on the thickness of paleosol sequences on Earth may be 9 km in paleosols of the Neogene Siwalik Group of India and Pakistan [1]. One hypothesis to explain the exceptional thickness of the Mawrth Vallis clays is that it may be a Fe/Mg-smectite paleosol sequence formed under a semi-arid climate, overlain by a subsequent, thinner, Al-rich leaching profile, and topped with an igneous caprock [23,24]. This stratigraphy is consistent with current climate hypotheses for early Mars, including a Noachian hyperarid, frigid paleoclimate, alternating with warmer and wetter conditions [14,26], or the possibility of a wetter but cold surface with a warm subsurface [50,51]. Additional evidence for widespread aqueous surface weathering on Mars has been provided by targeted CRISM observation of Al clays (~1 m thickness) overlying Fe/Mg smectite (>10 m thickness) clays nearValles Marineris, which could have formed from leaching by several hundred meters of highly acidic fluids [48]. More details may be forthcoming from the ExoMars 2020 rover landing at at Oxia Planum, which is a westward extension of the lowest portions of the Mawrth Vallis stratigraphy, and host to one of the largest clay units on Mars [23,52,53].

One feature of Mars that both in situ and orbital investigations observe is the widespread distribution of phyllosilicates (Figure 2). Phyllosilicates probably formed during transient warm and wet conditions during the Noachian (4.1–3.7 Ga), and are common across the surface of the planet [15,47,54]. Furthermore, these clays are not from deeply buried sedimentary rocks, because they show hydration and crystalline disorder which are incompatible with deep burial diagenesis [55].

On Earth, clay mineralogy is often determined by factors other than general temperature and water conditions, most notably the composition of the parent material. Soils derived from the weathering of ultramafic rocks like serpentinite can exhibit mixed di-trioctahedral clay composition [56]. Trioctahedral serpentine, although uncommon, has been detected on Mars [57], though trioctahedral clays have not been observed in layered vertical sequences [14]. Hence, potential paleosol sequences on Mars exhibit dominant spectral signatures of dioctahedral phyllosilicates in layered vertical profiles, and do not have signatures of trioctahedral phyllosilicates [22,46], suggesting that the parent materials of these potential paleosols may have lacked an ultramafic component. 

Two hypotheses to explain the formation and distribution of phyllosilicate clays on Mars are (1) subsurface hydrothermal activity and/or diagenesis [58,59,60], and (2) surface chemical weathering [10,14,23,61]. Hydrothermalism and/or diagenesis most often forms trioctahedral clay deposits exhibiting lateral variations in Al and Fe/Mg smectites intermixed with chlorite, serpentinite, talc and zeolite [14]. In contrast, dioctahedral smectite-rich outcrops with extensive vertical profiles of Al-smectites overlain by Fe/Mg smectites and then amorphous materials suggest subaerial formation in poorly-drained surface environments [14], consistent with pedogenesis. 

The study of fossil soils has historically drawn upon features of modern soils for the interpretation of paleosols and paleoenvironments, including the unique geochemistry of soils, soil horizons, soil structure and fossil root traces [1]. Because many paleosols are found in sedimentary rocks, and constitute alterations after deposition, the formation of paleosols in sedimentary rocks has traditionally been considered early diagenesis. This is distinct from late diagenesis, which is alteration after burial. Soils also are subject to alteration shortly after burial under low confining pressures, when they are still in communication with groundwater and air; this needs to be distinguished from deep burial diagenetic and metamorphic alterations; however, there appears to be an impressive lack of diagenetic maturation of chemical sediments across the surface of Mars [55]. Common diagenetic alterations to soils and their effects on indigenous organic matter preservation are outlined in the following paragraphs and provide guidelines for investigation of biosignatures in subaerial paleoenvironments of Mars.

### 1.2. Burial Decomposition of Organic Matter 

Many paleosols have significantly less organic carbon than their modern soil counterparts (Figure 3). The loss of paleosol organic carbon is facilitated by aerobic microbial decomposers that inhabit the overlying burial layer as part of the ecosystem forming soil on the sediment that buried the soil, or by facultative aerobic microbial communities which persist in the original soil after burial [1]. Both communities metabolize organic C to CO_2_, and are significant contributors to the decomposition of organic matter in paleosols. Other losses of organic matter may occur during deep burial and the generation of oil and gas, or after paleosol exhumation and exposure to photodissolution of organic matter [17], which, in modern soils, causes rapid losses up to several percent of soil organic matter in the top 10 cm over 10^1^–10^2^ yrs [62].

Burial decomposition of organic matter is especially obvious in Neogene (20–2.5 Ma) paleosols which are sufficiently similar to modern soils for comparison [1,63,64]. Despite the loss of organic matter, paleosols still show surface or shallow subsurface enrichment of organic carbon (Figure 3). Burial decomposition of organic matter is attenuated in soils that formed under reducing and waterlogged conditions, such as Permian (254 Ma) [65] and Oligocene (33 Ma) [66] coals, which are Histosols in the U.S Soil Taxonomy.

### 1.3. Illitization of Smectites

A common diagenetic alteration of paleosols is the burial-induced conversion of smectite to illite [67]. Diagenetic conditions that encourage illitization (1.2–3.2 km overburden and 55–100 °C) result in the loss of cations other than potassium (Ca^2+^, Mg^2+^, Na^+^) in expelled formation waters that would presumably also carry away organic matter. Illitization of smectite kinetics is partly controlled by the temperature, but also by availability, and thus, ion activity of potassium and ammonium, which are rate-limiting [68].

The crystal structure and surface charge of soil smectites encourages the formation of organo-mineral complexes [69], which are presumably altered or removed after illitization. Some paleosols may have originally been illitic in composition, like some desert soils of regions with strong seasonality effects, and lower productivity soils of the Paleozoic and Precambrian may have been illitic in composition [70]. However, mounting evidence from laboratory studies [71,72,73] and paleosol observations [70,74] suggest that illitization of smectite is common in the geological record.

It is commonly assumed that solid state-transformation or dissolution–recrystallization reactions (e.g., Ostwald ripening, the minimization of free energy by formation of progressively larger crystal faces) drive the smectite-illite conversion [71] but the presence of organo-mineral complexes, or microbes themselves, may influence illitization. The maturation of organic matter has been implicated in accelerating the conversion rate of smectite to illite by increasing the Gibbs free energy of illite growth [75]. Microbial mediation of illitization in dioctahedral smectites has been documented [73], and the desorption or decomposition of organic matter adsorbed in smectite interlayers at temperatures above 350 °C may control the degree of smectite illitization [71]. Similarly, common organic acids like acetate and oxalate can liberate Al and K ions to pore fluids by the dissolution of smectites, K-feldspar and muscovites [72]. The associated pH changes, along with increased amounts of K in pore solutions, have been documented to partially control smectite illitization in laboratory studies [71]. Thus organic carbon itself may influence the degree of alteration in paleosols subject to burial temperatures above 200 °C.

Certain clayey paleosols exhibit resistance to illitization of smectite. Clay-rich (>80 wt % smectite) Alfisols and Inceptisols from the Oligocene (33 Ma) middle Big Basin Member of the John Day formation in eastern Oregon [66] have resisted illitization despite burial by 1 km of overburden, likely aided by high clay content (Figure 4). Strikingly, the Inceptisols (Yellow paleosols, Figure 4) formed under reducing conditions have black iron manganese nodules (up to 1 m diameter) in subsurface (Bg) horizons, which are typical of seasonally waterlogged modern soils (Aquepts) and preserve higher amounts of organics relative to the bulk soil matrix. A lack of diagenetic illitization in these paleosols may be related to the preservation of these iron manganese nodules. Stratigraphically above, amorphous-rich (>30 wt %) and less clayey (<60 wt %) Aridisols and Andisols of the late Oligocene (26 Ma) Turtle Cove member show evidence of severe diagenetic alteration, including illitization of smectites [20] and celadonization from Ostwald ripening of an iron-rich smectite, exhibited by the striking lime-green color of these paleosols. These green paleosols exhibit more extensive burial illitization relative to older and more deeply buried paleosols, because they were Andisols with unstable, noncrystalline colloids from the weathering of vitric volcanic ash, unlike the smectite paleosols lower in the succession [66]. Therefore, certain clay-rich soils may be more resistant to diagenetic alteration relative to less clayey paleosols, even when buried by upwards of 1 km of overburden.

## 2. Type, Structure and Function of Organic Matter in Modern and Fossil Soils

A wide variety of organic compounds with variable structures, sizes, functions and reactivities have been documented in soils and paleosols. In modern soils, soil organic matter is a continuum of organic fragments that are progressively decayed by soil decomposers [76,77] which are then selectively preserved and altered after burial in paleosols. Progressive decreases in molecular size increase the number of polar and ionizable functional groups, thus increasing the solubility in water of soil organic matter [77]. Progressive decay to smaller molecules also encourages sorption of organic matter to mineral surfaces, which often leads to long-term recalcitrance [78]. However, many differences exist between the types of organic matter in soils and paleosols. The following paragraphs identify these differences and discuss implications for organic carbon preservation on Mars.

In the modern soil environment, a continuum of organic molecules exists which are variable in size, structure, function, reactivity, and resistance to degradation [77]. The largest of these pools is fragments from plants and microbial tissues which are predominantly composed of carbohydrate C, with lesser amounts of protein, lignin, and alkyl-C. The smallest and most “recalcitrant” are chemical structures dominated by alkyl-C [77]. Such recalcitrant compounds include long-chained *n*-alkanes with predominant odd-over-even chain length which are considered biomarkers of land plant leaf waxes [79].

Carbon from microbial biomass and/or metabolites can comprise a high proportion of the total organic carbon in paleosols. Biomarker ratios have been used to examine the proportion of C derived from plant versus microbial compounds in modern soils [80] and paleosols [78]. Quaternary (<2.5 Ma) paleosols, which typically have relatively low amounts of organic matter from plant biopolymers, are predominantly made up of N compounds and carbohydrates from microbial cells [81]. Biomarkers of chitin are abundant in Quaternary paleosols, a common cell wall component of insects and fungi, known from the presence of 3-acetamidofuran, 3-acetamide-5-methylfuran, 3-acetamido-2-pyrone and oxazoline compounds [81]. 

The clay-size fraction of modern soils [82] and Holocene paleosols [78] is dominated by aliphatic compounds from plant biomass, microbial biomass and/or metabolites, which are considered highly recalcitrant [83]. Holocene (~15 Ka) paleosols buried by loess in the Midwest of the United States have clay-size fractions with higher proportions of aliphatic compounds from microbial biomass (n-C_14_ alkanoic acid) relative to the overlying modern soil [78]. Biomarkers of the clay-sized fraction of much older Devonian (360 Ma) paleosols from Russia show organic carbon (0.5–5 wt %) composed mainly of carbon from aryls, acetyls and O-alkanes [64]. Selective preservation may be a result of paleosol mineralogy: smectite-dominated mineral matrices appear to preferentially preserve aliphatic structures compared to kaolinite- or palygorskite-dominated matrices [64]. The clay-size fraction of smectite-rich potential Martian paleosols thus offers the tantalizing prospect of being a favorable location for the retention of indigenous and highly recalcitrant organic carbon. However, exposure to ionizing radiation at the Martian surface and subsurface (0–500 cm) [36] can degrade organic carbon and other biosignatures like microbial carotenoids [84]. The duration of surface exposure and depth in the profile should be strongly considered when determining a sufficient location to sample the clay-sized fraction of potential Martian paleosols. 

Broadly, paleosols preserve highly recalcitrant organic carbon with a significant fraction derived from microbial biomass (e.g., iso-Alkanoic acids, [80]), but this does not include Histosols, which offer exceptional preservation of plant-derived organic matter as coal seams. More likely to be encountered on Noachean (4.1–3.7 Ga) terrains of Mars are Fe/Mg smectite-rich paleosol sequences with evidence of variable redox conditions [14], which are now considered a potential high-priority target for in situ biosignature detection [11] and Mars Sample Return [12,13]. However, it is important to note that exogenous organic carbon could have been delivered to the surface of early Mars by interplanetary dust particles, and could be a significant component of Noachian and pre-Noachian sediments [85].

## 3. The Nature of Organic Matter in Precambrian (>541 Ma) Paleosols

For the first two billion years of life on Earth, the main organisms contributing to the terrestrial organic carbon pool were microbes and microbial consortia, not plants. Microbes have left a record in organic carbon with a variety of life-like isotopic ratios in Precambrian paleosols [5,8,86,87], which indeed may be the best analogs for potential Martian paleosols, because they probably formed in the presence of microscopic organisms [88]. There are relatively few studies about the nature and composition of organic matter in Precambrian and older paleosols, and more research is warranted. However, it is clear that soil formation during the Precambrian probably involved a biological component, in contrast to speculation that Precambrian terrestrial weathering proceeded exclusively as an abiotic process [74]. 

Paleosols that formed after the evolution of land plants in the Ordovician (~470 Ma) were modernized by plant inputs. In modern soils, a significant proportion of plant organic matter passes through microbial biomass before becoming recalcitrant soil organic matter [89]. In any case, the organic matter in Precambrian paleosols is derived from a diverse suite of terrestrial microorganisms. Signatures of life in these ancient subaerial environments are known from the isotope ratios, biomarkers and/or microfossils of terrestrial organisms, including cyanobacteria [90], actinobacteria [5,91,92], methanogens [6] and fungi [93,94]. 

Precambrian (>541 Ma) paleosols on Earth have been subject to significant alteration after burial. Alterations include metamorphism [95], deformation [96], hydrothermal graphitization during serpentinization [8], illitization of smectite [67] and burial decomposition of organic matter [97]. Precambrian metamorphosed paleosols have been confused with upward-fining fluvial sequences, mudflow deposits, fault mylonites, marine hardgrounds, ash beds or zones of groundwater or hydrothermal alteration [5,70]. Fortunately, there is no evidence of metamorphic alteration to potential paleosols at the Gale Crater or possible sequences of paleosols at Mawrth Vallis on Mars [22,26,55]. Noachian surface paleoenvironments on Mars may therefore offer exceptional preservation of biosignatures.

## 4. Factors Influencing Organic Matter Preservation in Paleosols

### 4.1. Redox Chemistry

The most obvious control on organic matter preservation and degradation in paleosols is the redox state of the bulk soil prior to burial, especially evident from coals. Histosols are organic-rich soils with thick peaty horizons that typically form in low-lying, swampy and boggy areas with a shallow groundwater table, predominantly under anoxic and reducing conditions (Eh < −100 mV). Upon burial, these soils form coal seams, retaining organic carbon in amounts similar to the original soil (>25 wt % TOC), suggesting prolonged conditions that limit burial decomposition of organic matter. Microbial degradation is retarded under perpetually reducing conditions because anaerobic decomposition of organic carbon is much slower than aerobic decomposition. As is common with shallow or deep marine black shales, anoxic conditions of paleosols prior to burial provide a first order control on the preservation of organic matter.

Archean (2.3 Ga) paleosols with prominent reduced horizons from an anoxic atmosphere have organic matter concentrations ranging from 0.014–0.25% in the upper 2 m [74], possibly from microbial crusts [88,92], suggesting that clayey, poorly-drained soils preserve organic matter over geologic time scales [11]. Similar anoxic atmospheres may have been the rule for Mars over all of its known history [26], but the oxidation state of an early Mars atmosphere is not well constrained [28]. It is highly unlikely that paleosols as rich in organic matter as coals will be discovered on the surface of Mars, given satellite reconnaissance so far [46], but the discovery of carbonaceous layers in boreholes is possible. 

Molecular weathering ratios are a useful tool for estimating the redox state of a paleosol. The molecular ratio of ferrous to ferric iron (FeO/Fe_2_O_3_) can serve as a proxy for bulk redox state of a soil prior to burial [1]. Paleosols that formed under well-drained and strongly oxidizing conditions (Eh > +600 mV) have a characteristic FeO/Fe_2_O_3_ ratio of < 1. Variable redox conditions have FeO/Fe_2_O_3_ of ~ 0.5–1, while waterlogged soils of coal measures (Eh < −25 mV) have a FeO/Fe_2_O_3_ ratio of > 1, indicating prolonged anoxic conditions. These chemically reduced paleosols can often have orders of magnitude more organic carbon (TOC > 1 wt %) relative to oxidized paleosols. Soils of intermediate redox status (Eh ~ 0 mV) may have experienced partial water logging or periodic flooding [1]. The influence of a variable redox state on organic preservation in paleosols is not well understood, but it has been shown that certain types of organic matter can be preserved within the interlayers of smectite clay minerals against a changing redox environment [98]. Additionally, modern Ultisols with prominent redoximorphoic features exhibit depth-dependent crystallinity of iron oxides [99], which may influence organic matter preservation and degradation. Lower crystallinity phases are generally more reactive as sorbents for organic compounds and electron acceptors for organic matter mineralization relative to strongly crystalline iron oxides [99]. Certain intermediate-redox state paleosols exhibit strong redoximorphic features like mottling (spots or blotches of different color), probably from the hydrolytic liberation of drab-colored (dull olive-green) Fe (II) during reducing conditions which, upon seasonal drying of the soil, oxidize to Fe (III) hydroxides including goethite. A compilation of FeO/Fe_2_O_3_ of Permian (260 Ma) paleosols from Antarctica show a significant (R^2^ = 0.6, *P* < 0.0001) relationship between redox state and total organic carbon preserved in the soil profile (Figure 5).

It is broadly assumed that clay minerals can aid in the preservation of organics in redox-active soils. However, the oxidation state of iron in clay minerals may play a key role in the preservation of organics in clayey paleosols of variable redox state. The Fe (III) reducing bacteria *Shewanella putrefaciens* can liberate intercalated organic matter upon reductive dissolution of small and poorly crystalline phases of nontronite, a Fe (III) dioctahedral smectite clay whose formation is typically associated with anoxic conditions, but this does not occur in larger and more crystalline nontronite phases [98]. Variable redox state may indeed encourage the preservation of organic matter in strongly crystalline Fe/Mg smectites, which appear to be common across Noachian (4.1–3.7 Ga) subaerial paleoenvironments of Mars [24]. Strongly crystalline nontronite, however, has been experimentally shown to form under highly oxidizing (>800 mV) conditions from solutions containing Fe (III) and Mg (II) under Mars-like temperatures and pressures [100]. Strongly oxidizing conditions do not typically favor the preservation of organics intercalated in the interlayer spaces or on mineral surfaces of smectites, so the presence of dioctahedral smectites alone may not necessarily bring about high organic preservation potential. Instead, reliance on discreet molecular weathering ratios like FeO/Fe_2_O_3_ can be used to target high-priority locations for biosignature investigation in the ancient subaerial paleoenvironments of Mars. 

### 4.2. Depth

Similar to modern soils, many paleosols exhibit a surface enrichment of organic matter relative to lower horizons [8]. Depth functions for organic carbon content are variable, but are at least one order of magnitude lower in Precambrian (>541 Ma) paleosols when compared with modern or Pleistocene paleosols, probably due to the extended burial decomposition of organic matter and/or diagenesis. Despite the presumed low biomass of Precambrian organisms per soil volume unit, even the most ancient putative soil organisms were likely concentrated in surface horizons [74,88].

Analysis of total organic carbon from Eocene (33 Ma) paleosols of the John Day Fossil Beds National Monument in eastern Oregon showed a marked surface enrichment of organic carbon in two out of three profiles examined (Figure 6). Although these soils formed under oxidizing conditions, and most indigenous organic matter had been removed, the surface horizons of the uppermost soil had the highest amounts of organic carbon of any location in any of the three paleosols. Thus, the depth at which to sample is an important feature to consider when prioritizing locations for biosignature investigation on Mars. When faced with a sequence of bedded rocks overlying gradationally altered soil below, the samples most likely to be rich in organic matter will be just below the upper contact.

### 4.3. Clay Mineralogy

Relationships between organic matter content and mineral surface area have been widely reported across terrestrial and marine settings. Mineral surface area and cation exchange capacity are dictated by clay mineralogy, so natural differences arise between different types of clays and their affinities to form organo-mineral complexes. However, the influence of clay mineralogy on the stability and persistence of organo-mineral complexes through deep time is poorly understood [101]. Compounds are well-preserved in the interlayer spaces of certain phyllosilicate clay minerals because the charged surface area of these minerals sorbs and retains organic matter [102,103] and can be extremely resistant to desorption [104], making the separation of organics and clays rather difficult. Interlayer binding acts as a retention mechanism for the sorption of organic compounds [105], including amino acids onto phyllosilicates [106], which may shelter organic matter from oxidation [98] and radiation exposure [14,59]. 

The surfaces of soil clay minerals are favorable locations for organic carbon preservation, and several pathways have been identified which can preserve organic molecules through geologic time. These include the occlusion (physical protection) of organic matter, organo-mineral sorption to phyllosilicate surfaces, crystal edges, and interlayer spaces, and/or the establishment of chemically reducing microenvironments that shelter organic matter from degradation [107]. Mineral sorption is especially effective in fixing labile soluble organic matter, including amino acids, carbohydrates, fatty acids and RNA oligomers.

Clay mineralogy is generally reflected in the degree of weathering of a soil. A high kaolinite content of soils and paleosols is characteristic of highly weathered soils including Ultisols and Oxisols, which accumulate Al and Fe at the expense of soluble cations as a result of sustained hydrolytic weathering (Figure 7). Smectites are a group of 2:1 clay minerals which are common in moderately weathered soils and paleosols, and are thought to be more favorable for the preservation of organics relative to 1:1 kaolinitic clay minerals, discussed below.

In clayey (>80 wt % clay) paleosols, the molecular ratio of bases to alumina (CaO + MgO/Al_2_O_3_) is a proxy for smectite to kaolinite clay mineralogy [1]. Values of > 1 for bases/alumina suggest smectite as the principal clay mineral, while low (<0.5) values (increasing Al_2_O_3_) indicate increasing kaolinite content. However, it is not suitable to apply this proxy for less clayey soils (<60 wt %) because values can be compromised if there are still a significant fraction of unweathered primary minerals, characteristic of less mature and thus less clayey soils. A compilation of Cenozoic paleosols with greater than 60 wt % clay shows that organic C is related to the molecular ratio of CaO + MgO/Al_2_O_3_ (Figure 8). Early Eocene (55 Ma) clay-rich acid-sulfate paleosols from Big Bend National Park, Texas show chemical index of alteration (CIA) ranging from 78–80, indicating extreme weathering conditions and resulting in kaolinite as the principal clay mineral [109]. Extreme weathering is characteristic of Ultisols, which form in humid, tropical settings [110]. These kaolinite-bearing Ultisols have bases/alumina ratios of 0.08–0.1 and low (<0.2 wt %) TOC (Figure 8). In contrast, less-weathered Miocene (8.5 Ma) Alfisols (clayey, forest soils [110]) from Pakistan [111] have bases/alumina of ~ 0.3–0.8 and higher TOC values (0.22–0.8 wt %, Figure 8), which reflects the smectite composition and moderate weathering during soil formation. Considerably less-weathered Miocene (8.5 Ma) Inceptisols also from the Siwaluk formation in Pakistan show bases/alumina of > 0.6 and the highest TOC (0.1–0.3 wt %). These soils probably formed in perennially saturated conditions because they are classified as “aquic Inceptisols” [110], implying groundwater presence within 100 cm of the soil surface for some part of the year. Inceptisols that formed under perennially saturated conditions often have higher TOC relative to well-drained Inceptisols of similar mineralogy. As such, it is important to consider clay mineralogy in conjunction with redox state (Figure 6).

Smectites have the highest mineral surface area and cation exchange capacity of all phyllosilicates [112], so they are broadly considered a favorable clay mineral group for the preservation of organo-mineral complexes and other biosignatures. However, there are many factors that influence organo-mineral stability and persistence, including pH-dependent surface charge and the ionic strength of pore waters. For example, pedogenic smectites can be stripped of their organic carbon upon deposition in distal marine settings, probably due to changes in ionic composition and strength once submerged in seawater [101]. 

Wattel-Koekkoek et al. [104] studied differences in organic matter residence time in modern pedogenic smectite versus kaolinite, and found large differences in the residence time of organic C in kaolinite (360 years) relative to smectite (1100 years), concluding that clay mineralogy is the main factor explaining differences in extracted organic C levels (Table 1). 

This may be a result of cationic bridging of organic matter in smectite, not thought to be common in organic matter associated with kaolinite, where, instead, organics are free or sorbed to the aluminum hydroxide surface of kaolinite [69]. Thus soils rich in original smectite clays may retain more organic matter than paleosols rich in kaolin clays because of differences in chemical bonding between organic carbon and mineral surfaces. Fortunately, dioctahedral Fe/Mg smectite clays have been detected from orbit at Jezero Crater, the final landing site for the Mars 2020 mission [113] and at Oxia Planum, the landing site of ExoMars 2020 mission [114].

### 4.4. Clay Abundance

The weight percent of clay in a paleosol has also been shown to influence organic matter preservation and susceptibility to diagenesis [10,66]. Paleosols with high (>80 wt %) abundances of smectite clays generally have higher amounts of organic carbon and less diagenetic alteration relative to less clayey paleosols [66]. Importantly, paleosols with high smectite clay content and low abundances of amorphous colloids exhibit resistance to severe diagenetic alteration, including illitization, celadonization and zeolitization [10]. This is most noticeable within Mars-relevant paleosols at the John Day Fossil Beds National Monument in eastern Oregon, where less clayey Andisols and Aridisols rich in amorphous colloids (imogolite, allophane) from the late Oligocene (26 Ma) exhibit severe diagenetic alteration relative to older, more developed and clay-rich paleosols (Alfisols and Ultisols) of the early Oligocene (33 Ma), despite greater depths of burial.

Soils and sediments share some characteristics related to the storage of organic matter, despite profound differences in the way clay minerals are delivered, transformed and interact with organics in each system [112]. Detrital minerals in marine sediments are often delivered from the sedimentation of terrestrial mineral assemblages, and a significant amount of shallow marine organic carbon can be pedogenic in origin [77], often depending on local uplift rate [101]. Within wetland (saturated) soils and marine environments, concentrations of organic carbon are correlated with the abundance of clay minerals [76,112,115], although temperature and water availability create increased variance in soil environments [115]. In formerly well drained paleosols, however, even clayey profiles may have little organic matter because of decomposition by aerobic microbes early after burial (Figure 3).

Paleosols with higher clay content generally have higher total organic carbon. A compilation of total organic carbon content of five Miocene (16–20 Ma) paleosols from Australia, Pakistan and Kenya showed that the organic matter content in each soil is related to the percentage of clay (Figure 9). This relationship may be due to the high specific surface area and cation exchange capacity of clayey soils providing more surfaces for the formation of stable organo-mineral complexes, but the redox state of the soil during formation is probably more important for organic carbon preservation. For example, clay-rich tropical rainforest Oxisols have abundant kaolinite clay, but only trace (<0.1 wt %) amounts of organic carbon, while less-weathered smectitic paleosols (Alfisols, Inceptisols) preserve much more (>1 wt. %) organic carbon (Appendix A). Therefore, clay abundance alone may not prove useful for predicting organic carbon, but may be more useful for characterizing subaeriel paleoenvironments with high preservation potential when other factors like redox state and clay mineralogy are considered.

### 4.5. Composition and Abundance of Amorphous Materials

Amorphous phases are formed from primary (e.g., volcanism, impacts) or secondary processes (e.g., pedogenesis, hydrothermal alteration, radiation damage) [116,117]. Amorphous colloidal noncrystalline phases are abundant in soils when instantaneous weathering rates are high and/or when parent materials are high in amorphous materials, such as volcanic glass [118]. Amorphous materials are a common product of volcanic soils and consist mainly of hydrolyzed Al, Si and Fe; these materials are characterized by high specific surface area and reactivity. The chemical weathering of volcanic ash begins with the leaching of soluble components (desilication) including H_4_SiO_4,_ Ca^2+^, Mg^2+^, Na^+^ and K^+^, and is accelerated by the presence of carbonic acid, resulting in the accumulation of sesquioxides (e.g., Al_2_O_3_) and the formation of secondary amorphous minerals from aluminum and silicic acid. In many volcaniclastic soils, amorphous ferrihydrite ripens first to magnetite and/or maghemite, and then ultimately to hematite [119]. The composition of amorphous materials in volcanic soils is also a function of soil permeability; if soils are well drained, soluble reaction products are removed rapidly from leaching, whereas poorly drained soils accumulate soluble reaction products including Mg^2+^ and, in some cases, form poorly crystalline smectites including montmorillonite ([119]. 

Reactive amorphous phases of Al and Fe oxyhydroxides in soils adsorb and retain organic matter [49,102]. Amorphous Al(OH)_3_ is known to have a strong affinity to chemisorb organic matter, and thus, to provide a strong preservation mechanism in soils [119,120,121]. Poorly crystalline Fe (oxy)hydroxides are generally more reactive than higher crystallinity phases as sorbents for organic matter [99]. In modern soils, ligand exchange between poorly crystalline mineral surface hydroxyl groups and negatively charged organic functional groups can stabilize and preserve organic matter in subsurface horizons of acid soils [102]. DNA from Holocene volcaniclastic paleosols (Andisols) is often adsorbed, chemisorbed and/or encased in nanopores of amorphous allophane which can be successfully extracted and amplified using wet chemistry methods [122]. Amorphous materials may therefore serve as a repository for organic matter in paleosols, especially those lacking evidence of severe diagenetic alteration.

X-ray amorphous phases have been detected (approximately 15–70 wt. %) in all soil and rock samples analyzed by the CheMin x-ray diffractometer at Mars Science Laboratory (MSL) onboard Curiosity rover [123]; however, the composition of these amorphous phases is difficult to constrain, and may range from primary (volcanic glass) to secondary (silica, Fe [oxy]hydroxides) phases alone or in combination [118]. Approaches to resolve the composition of amorphous phases at the Gale Crater have employed a combination of bulk mineral abundances from CheMin and bulk chemical composition from the Alpha Particle X-ray Spectrometer (AXPS), also onboard MSL. X-ray diffraction refinements, pattern fitting and mass-balance calculations applied to terrestrial samples reveal compositional similarities between certain Martian soils and sediments and modern glacial sediments, volcanic soils and volcaniclastic paleosols [124]. Rhyodacitic to andesitic paleosols and modern glacial sediments show abundances of amorphous SiO_2_, TiO_2_ and Al_2_O_3_ phases similar to those of mudstones at Gale Crater [124].

Amorphous phases in volcaniclastic paleosols commonly include volcanic glass, allophane (Al_2_O_3_· [SiO_2_]_1.3–2_·2.5–3 H_2_O), nanocrystalline silica and imogolite (Al_2_SiO_3_[OH]_4_). The composition of amorphous phases in vitric tuffaceous soils and paleosols is dominated by aluminum oxides and silicates. These distinctive soils, called Andisols, often have a dark organic surface horizon, known as a melanic epipedon in US Soil Taxonomy. In such soils, the decomposition of organic matter is hindered by sorption to amorphous aluminum hydroxides. As these soils mature, the increasing crystallinity of amorphous phases in modern Andisols leads to decreased amounts of TOC (Figure 10). The “activity ratio” [119] of active iron oxides to free iron oxides, determined by Mossbauer spectroscopy and expressed in weight percent, is a proxy for the degree of crystallinity of amorphous soil minerals, and is significantly correlated with the total organic carbon in modern soils (Figure 10). Ultisols are much more developed soils relative to Andisols with mineralogy dominated by crystalline compounds and lower values of active iron oxides/free iron oxides (0.15–0.32) than Andisols (0.30–0.99) [119]. 

Volcaniclastic paleosols formed in volcanic glass are often dominated by imogolite, which can—through diagenetic Ostwald ripening—transform to celadonite or clinoptilolite [66,118]. These paleosols, however, are unusually barren of organic matter, presumably lost in water expelled by diagenetic alterations resulting from deep burial. Such deep burial diagenetic alterations have not been observed in Martian clays so far [55]. Together, the large specific surface area, high reactivity and lack of diagenetic alteration suggest that amorphous-rich volcaniclastic paleosols on Mars are a high-priority target for biosignature investigation [11] and Mars Sample Return [13].

### 4.6. Sulfur Aids Preservation of Organic Matter

Interactions between organic matter and sulfur can encourage the long-term (10^6–^10^9^ yr) preservation of organic matter in sedimentary rocks [107,125,126,127,128] and paleosols [126]. Sulfurization is the incorporation of reduced inorganic sulfur species into sedimentary organic matter [129], and has been cited as a dominant preservation mechanism for functionalized organic compounds during early diagenesis [130]. The addition of sulfur encourages structural cross-linking of organic components, creating a macromolecular structure [128] which attenuates the degradation of organics by the addition of an oxidative sink [107], and possibly results in organo-sulfur compounds which are incompatible with exoenzyme degradation [130]. Organic carbon detected at the base of the Murray formation at the Gale Grater is composed of methanethiol, dimethylsulfide, carbonyl sulfide and carbon disulfide [107], which are common products of early diagenetic sulfurization reactions involving organic carbon. These compounds indicate that the diagenetic alteration of rocks and organics with acidic fluids in the lower Murray formation [123] was not sufficient to oxidize and remove all organic carbon [107]. 

Similarly, terrestrial sulfurization reactions documented in Archean rocks has been shown to enhance the short-term preservation of organic carbon while encouraging long-term resistance to acid diagenesis, oxidation and cosmic ray-induced degradation [107]. Potential paleosols of the Archean (3.7–2.5 Ga) age show geochemical evidence of formation under an anoxic, reducing atmosphere [5,6,74], even when formed in well-drained settings [95]. Furthermore, many Archean paleosols have abundant sulfates including gypsum, barite and perhaps kieserite [5,87], associated with other evidence of sulfur oxidizing photosynthetic bacteria [92]. Geologically younger paleosols from the upper Triassic (215 Ma) Chinle formation in Arizona [131] contain the sulfate mineral jarosite, which appears to encourage organic matter preservation when forming a rind on calcite-rich nodules [132], while organosulfur compounds including thiophenes and thienothiophenes are common in late Jurassic paleosols from the “dirt beds” near Dorset, UK [126]. Considering these large temporal scales across terrestrial environments, the role of sulfur in the preservation of organic compounds may be a common phenomenon throughout the fossil record of soils.

Sulfur may have played a role in the preservation of organic carbon in potential Archean (3.0 Ga) alluvial paleosols in the Farrel Quartzite of Western Australia (Figure 11) [92]. However, the presence of paleosols and the nonmarine affinity for carbonaceous microfossils has been disputed [133]. Several lines of evidence support a pedogenic and nonmarine hypothesis for the formation of the Farrel Quartzite, as summarized in [92]. Most notably, the presence of replacive barite (BaSO_4_) sand crystals and silicified pseudomorphs of nahcolite (NaHCO_3_) grouped into distinct layers resembling a soil gypsic (By) horizon [110] are thought to be associated with subaerial formation environments [92]. Alternatively, this unit has been interpreted as a microfossil-bearing carbonaceous chert deposited in a shallow marine to subaeriel sedimentary setting [134], and accordingly, there is an ongoing debate about whether the paleosol hypothesis is correct [133]. From a soil science perspective, the subaerial exposure of marine sediments and the formation of evaporitic sulfate minerals implies extended subaeriel chemical weathering, and allows for classification as Gypsid soils (salt crusts) in USDA soil Taxonomy [110], though the sand-sized barite nodules described as pedogenic features could be reworked grains of hydrothermal barite [133]. Despite disagreement on the origin of the Farrel Quartzite, the influence of sulfur on the preservation of organic carbon in these rocks requires examination, as discussed below.

In what has been interpreted as the gypsic (By) horizon, abundant pyrite (FeS) framboids were associated with spheroidal and spindle-like carbonaceous microfossils, which were putatively identified as *Archaeosphaeroides* and *Eopoikilofusa,* respectively, possibly representing a terrestrial consortium of actinobactera, purple sulfur bacteria and methanogens [92]. In contrast, carbonaceous microfossils in the Farrel Quartzite were first described as aquatic organisms based on stratigraphic, petrographic and trace element evidence [135]. Though the affinities remain disputed, there is general agreement that they are putative microfossils and not pseudofossils [136], which is partially supported by their association with sulfur-bearing minerals including pyrite framboids [92]. Pyrite framboids are subspherical to spherical clusters of submicron to micron-sized pyrite crystals packed together in tight association [137], resembling the aggregate of druplets of a raspberry. Photosynthetic sulfur bacteria may have been responsible for the accumulation of barite, which then served as the electron acceptor for endolithic sulfate-reducing bacteria whose metabolic activity is represented by clusters of opaque pyrite framboids [92]. Opaque pyrite framboids can represent traces of the activity of sulfur-reducing microbes [138] because they are known to grow in association with modern anaerobic microbial biofilms [137]. 

The abundance of pyrite framboids is related to the organic carbon content of potential paleosols in the Farrel Quartzite (Figure 11) because gypsic (By) horizons with abundant (>500) pyrite framboids were found to contain higher levels of total organic carbon relative to those with lower (<100) abundances (Figure 12, Appendix A). It should be noted that only eleven samples were analyzed for this work, and thus, the data point with the highest TOC (0.12 wt. %) and greatest number of pyrite framboids (866 framboids) (sample R4339, Figure 11) largely controlled the slope and significance of the relationship (Appendix A). Nevertheless these data suggest that subaeriel or shallow marine Archean rocks with evaporitic sulfate minerals and/or framboidal pyrite can be conducive to the preservation of organics associated with microbial metabolism, and should be considered a high-priority target for biosignature investigation in sulfur-rich subaerial paleoenvironments on Mars [132].

## 5. Conclusions

Ancient soils are subject to a wide range of physical, chemical and biological processes that facilitate organic carbon preservation and degradation over geological time scales. The results of a global compilation of organic matter content of soils and paleosols showed that the redox state of a paleosol prior to burial is the predominant control on the preservation of organic matter. Paleosols that formed under oxidizing conditions have significantly lower total organic carbon prior to burial, and are more prone to burial decomposition of organic matter than poorly-drained paleosols that formed under reducing conditions. However, diagenetic alteration, depth in the soil profile, clay mineralogy, amount of amorphous colloids and sulfur content are all important factors that influence organic matter preservation and degradation in paleosols Notably, Oligocene (33 Ma) Mars-relevant paleosols with abundant (>80 wt %) dioctahedral Fe/Mg smectite clays appear to resist severe diagenetic alteration including illitization and celadonization, which may liberate mineral-associated organic carbon from paleosols with lower clay content and higher amounts of amorphous colloids. 

Severe diagenetic alterations of Earth’s oldest putative soils have transformed original clay minerals and amorphous colloids, and only pseudomorphs of sulfur-bearing evaporite minerals appear to remain, though pyrite framboids persist. Despite the diagenetic burial decomposition of organic matter, many Phanerozoic (<541 Ma) and Precambrian (>541 Ma) paleosols exhibit surface and shallow subsurface enrichment of organic carbon, similar to modern soils. Additionally, paleosols with smectite clays are more favorable for the preservation of organic carbon relative to paleosols with accumulations of kaolin clays. Importantly, the clay-size fraction of modern soils and paleosols can preserve highly recalcitrant organic matter derived from microbial biomass and/or metabolites, which suggests that clay-size fractions of smectite-rich potential Martian paleosols should be prioritized for biosignature investigation. Data also show that amorphous colloids, common in soils and paleosols, are favorable locations for the sorption of organic carbon, even though the diagenetic alteration of amorphous-rich paleosols can remove organic compounds. Finally, potential Archean (3.0 Ga) acid sulfate paleosols show evidence of biomineralized pyrite (FeS) and the possible persistence of organo-sulfur compounds because horizons with accumulations of framboidal pyrite have significantly higher total organic carbon relative to those with lower accumulations of framboidal pyrite. 

A summary of organic matter content of paleosols spanning the geological record on Earth showed that organic carbon is most abundant in the upper horizons of paleosols that are unoxidized, rich in smectites, amorphous colloids and sulfur. Future approaches to biosignature detection in possible paleosols on Mars should target regions and horizons that fit these criteria. The factors presented here are all aspects of soil science; therefore, soil scientists should continue to be involved in the current and future exploration of Mars.

## Figures and Tables

**Figure 1 life-10-00113-f001:**
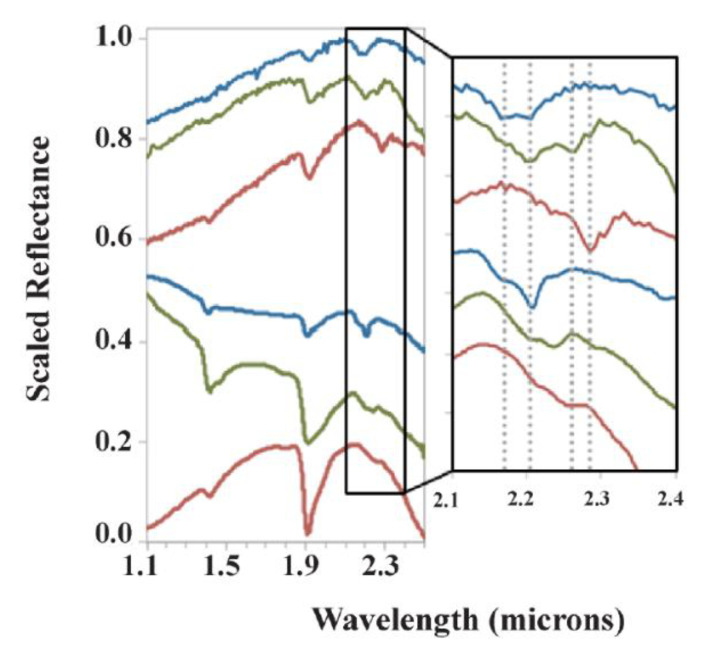
Spectral comparisons of minerals in terrestrial and Martian subaerial paleoenvironments. Compact Reconnaissance Spectrometer for Mars (CRISM) spectra (FRT863E) from Mawrth Vallis compared with laboratory spectra of a volcaniclastic paleosol from the Oligocene (33 Ma) John Day formation in Oregon with enlarged area to right (adapted from [20]). Top three: CRISM kaolinite, Al/Si doublet, and nontronite dominated spectra. Bottom three: John Day paleosol spectra with varying contributions of kaolins, illite, smectites and silica, as inferred from both near and thermal IR spectra (adapted from [20]).

**Figure 2 life-10-00113-f002:**
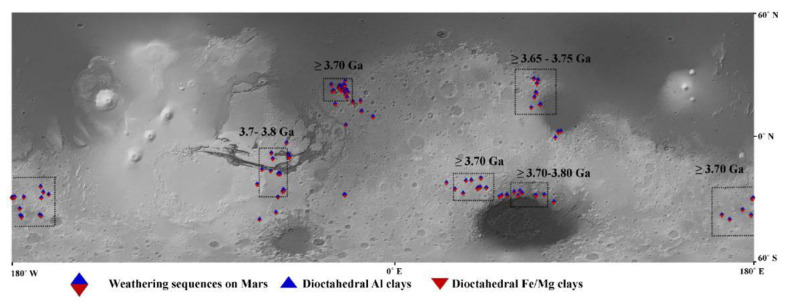
Vertical stratigraphies of dioctahedral Al overlying Fe/Mg clays from CRISM spectra, indicating possible weathering sequences or paleosols (e.g., Al clays stratigraphically above Fe/Mg clays and never in the opposite sequence) across the surface of Mars, with boxes indicating the regions for which the likely age of weathering is constrained (adapted from [23]).

**Figure 3 life-10-00113-f003:**
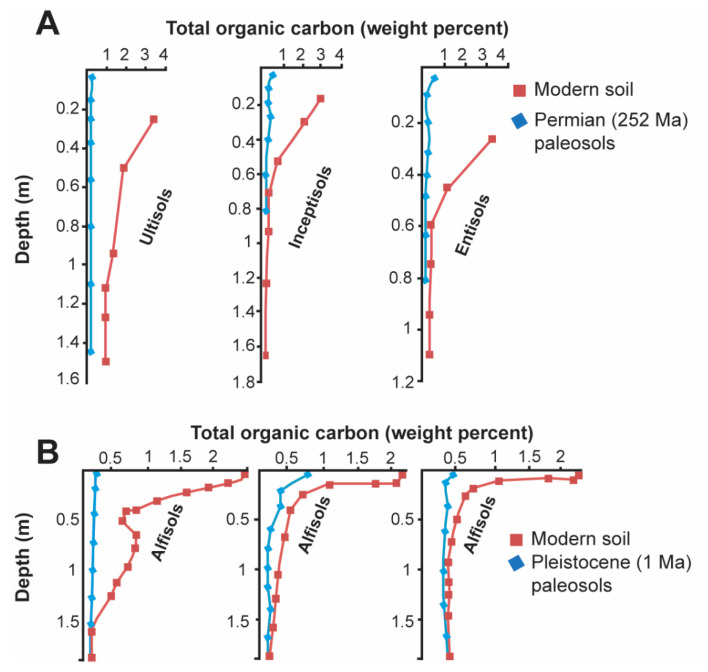
Depth profiles showing burial decomposition of organic matter in Permian (**A**) and Pleistocene (**B**) paleosols, and comparisons to modern soils. (**A**) Permian (252 Ma) paleosols from Graphite Peak, Antarctica (blue lines) and comparison with modern soil equivalents (red lines); (**B**) Pleistocene (0.2–1 Ma) paleosols (blue lines) compared to overlying modern soils (red lines) from the Midwest United States (Adapted from [1]).

**Figure 4 life-10-00113-f004:**
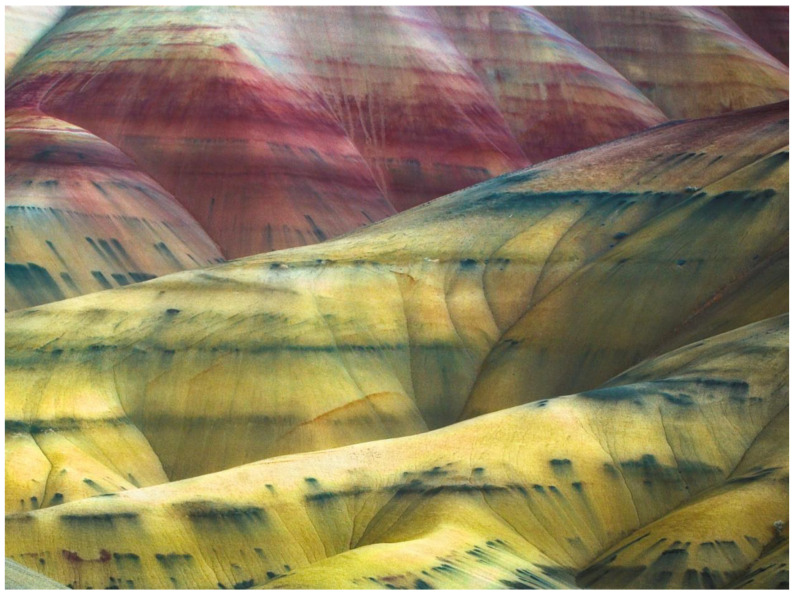
Early Oligocene (33 Ma) Mars-relevant paleosols in the middle Big Basin Member of the John Day Formation, eastern Oregon, USA. Repeating sequences of yellow clayey soils (Inceptisols) rich in Al-smectite formed under chemically reducing conditions and have horizons of abundant large (up to 1 m diameter) black iron manganese nodules associated with organic matter, while red soils (Alfisols) with abundant Al smectite and Fe oxides formed under oxidizing, well-drained conditions. Both paleosols exhibit minimal or absent illitization of smectite.

**Figure 5 life-10-00113-f005:**
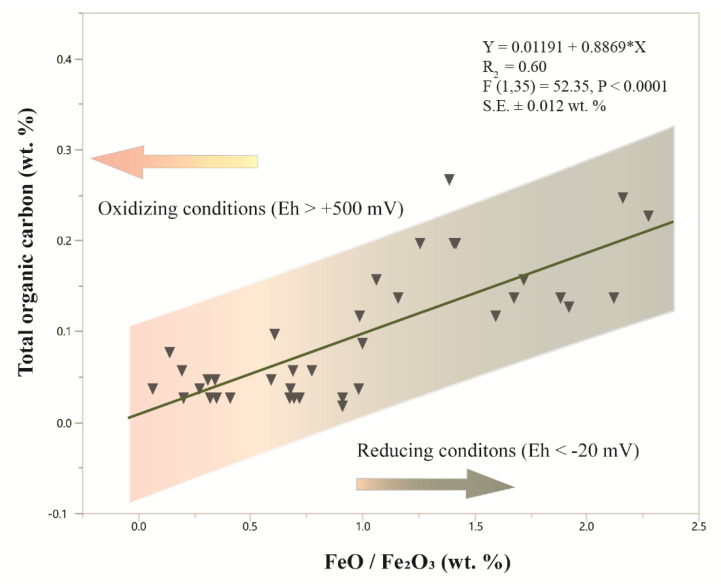
Relationship between total organic carbon and the molecular ratio of FeO/Fe_2_O_3_ in Permian (260 Ma) and Triassic (251) paleosols from Antarctica. Values of FeO/Fe_2_O_3_ greater than one generally suggest reducing conditions during soil formation; values less than one are associated with oxidizing conditions during soil formation. Data are from [65], and included as Appendix A.

**Figure 6 life-10-00113-f006:**
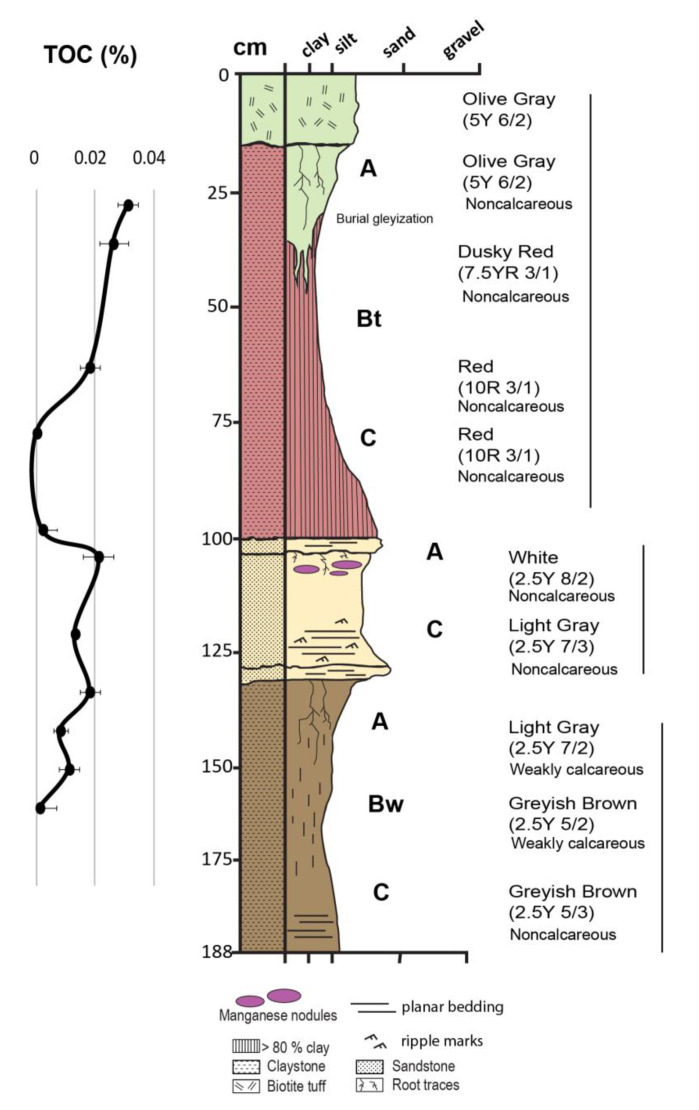
Total organic carbon (TOC), lithology and morphology of three paleosols from the John Day Fossil Beds National Monument, Oregon, USA showing surface enrichment of organic carbon. Error bars (*n* = 2) were constructed using one standard deviation from the mean.

**Figure 7 life-10-00113-f007:**
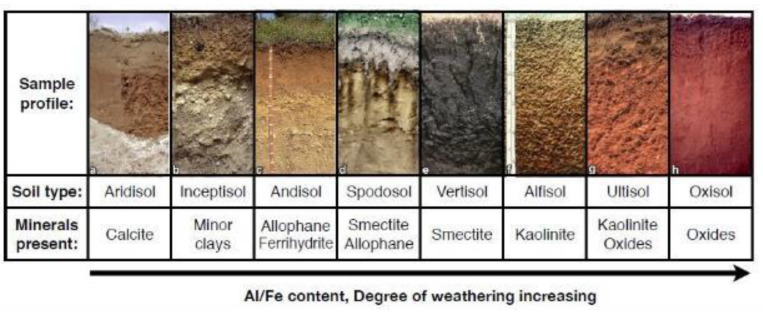
Dominant mineralogy of eight modern soil types (A-H) as a function of weathering. Minimally weathered soils include Aridisols and Inceptisols, while highly weathered soils include Ultisols and Alfisols (From [108], used with permission.).

**Figure 8 life-10-00113-f008:**
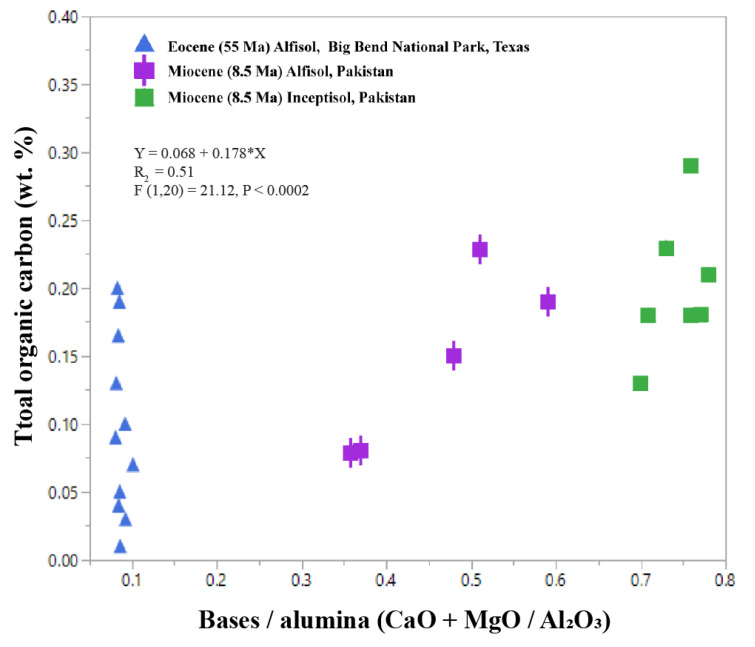
Total organic carbon (wt %) versus the molecular ratio of bases to alumina (CaO + MgO/Al_2_O_3_, wt %) in Cenozoic (55–8.5 Ma) clayey paleosols. Bases/alumina is a proxy for kaolinite to smectite clay mineralogy of clay-rich > 60 wt % paleosols: values of < 0.1 suggest kaolinite and values of > 0.1 suggest smectite mineralogy.

**Figure 9 life-10-00113-f009:**
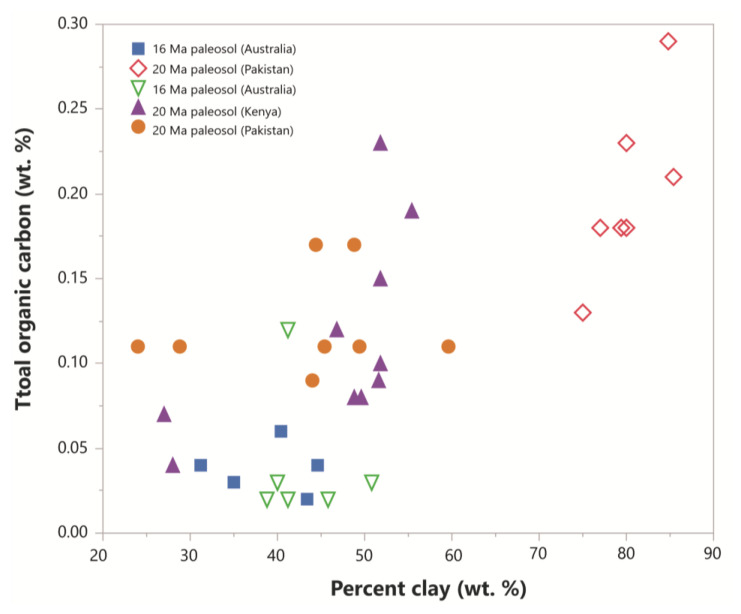
The relationship between clay content and total organic carbon in Miocene paleosols (all Alfisols) from Australia (16–20 Ma), Pakistan (20 Ma) and Kenya (20 Ma). Data are from Metzger and Retallack, 2010, and included as Appendix A.

**Figure 10 life-10-00113-f010:**
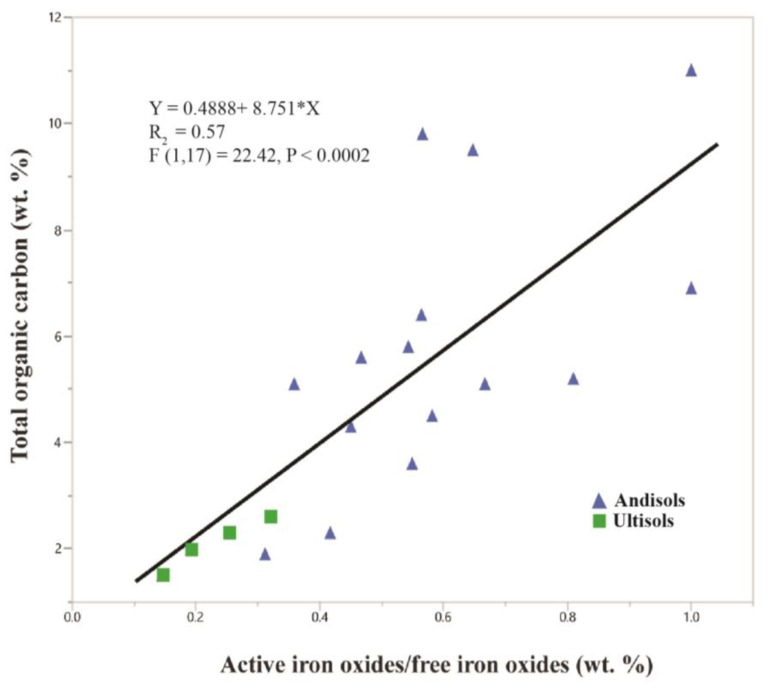
Total organic carbon versus the ratio of active to free iron oxides in modern Andisols (amorphous-rich volcanic soils, blue triangles) and Ultisols (highly weathered, highly crystalline soils, green squares) from Chile. The ratio of active to free iron oxides is a proxy for Fe_2_O_3_ crystallinity; increasing values of active to free Al_2_O_3_ are indicative of higher amounts of amorphous Fe_2_O_3_ (Data from [119]).

**Figure 11 life-10-00113-f011:**
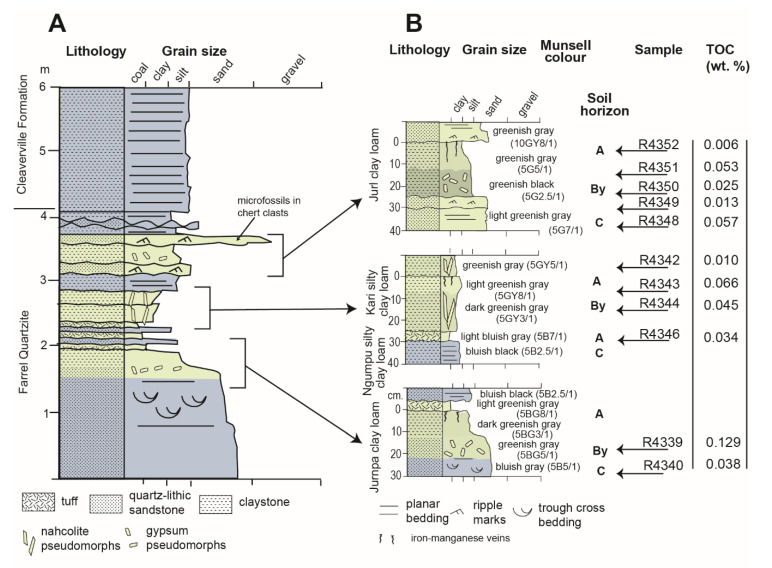
Stratigraphic section (**A**) and potential paleosol profiles (**B**) from the 3.0 Ga Farrel Quartzite, Western Australia (redrafted from [92]). “Sample” in (**B**) correspond to locations where analysis of total organic carbon (TOC) was performed on hand samples previously used to collect oriented thin sections for the determination of framboidal pyrite abundance [92]. Determination of TOC was performed by combustion on a Costech ECS 4010 elemental analyzer. Samples were pretreated with HCl to remove inorganic carbonates. The total organic carbon values displayed are the averages of two replicates. All samples are in the Condon Collection of the Museum of Natural and Cultural History of the University of Oregon. All data are included as Appendix A.

**Figure 12 life-10-00113-f012:**
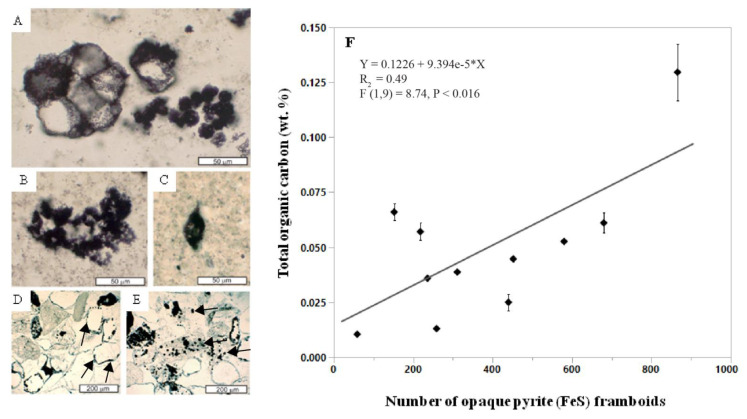
Microfossils and opaque pyrite framboids from Archean (3.0 Ga) potential paleosols in the Farrel Quartzite, Western Australia, and the relationship between total organic carbon and pyrite framboids. (**A**) clusters of large thick wall spheroid (left: some deformed by crystal growth) and small spheroid (right) morphotypes; (**B**) cluster of small spheroid morphotype; (**C**) spindle morphotype; (**D**,**E**) sandstone with grains of fossiliferous chert (grey) and quartz (white), including marginal opaque framboids of possible endolithic microbes (black arrows); (**F**) Least-squares regression of mean total organic carbon content (TOC) versus the number of opaque pyrite framboids in the clasts and matrix of an Archean (3.0 Ga) coastal-plain paleosol in the Farrel Quartzite, Western Australia. Error bars are constructed using one standard error from the mean. Hand samples and oriented thin sections in the Condon Collection of the Museum of Natural and Cultural History of the University of Oregon are (A–C) F118310B = R4336, (D–E) F118310E = R4339. All data are included as Appendix A.

**Table 1 life-10-00113-t001:** Mean age of organic carbon in modern kaolinitic and smectitc soils (data from [104].)

Site	MAT (C)	Rainfall (mm/yr)	Soil Type ^φ^	Clay (g/kg)	TOC (g/kg) ^Ʊ^	Kaolinite (%) ^†^	Mica (%)	Smectite (%)	Mean Age of Org. C (years) ^‡^
**Kaolinitic**									
Brazil	23	1317	Haplic Lixisol	150	10	92	8	0	187
Brazil	23	1234	Haplic Lixisol	260	10	98	2	0	487
Mali	27	1087	Ferric Acrisol	230	9	96	4	0	162
Mozam-bique	24	932	Ferric Acrisol	270	23	100	0	0	545
**Smectitic**									
Kenya	18	504	Pellitic Vertisol	600	14	1	0	99	1368
Nicaragua	27	1184	Pellitic Vertisol	790	11	0	0	100	629
South Africa	19	928	Pellitic Vertisol	460	24	21	0	79	903
South Africa	19	928	Pellitic Vertisol	570	16	21	0	79	1302

φ Soil type is after USDA Keys to Soil Taxonomy, 2014; † Clay mineralogy determined by XRD and displayed as relative peak area of diffractograms; ‡ Age of organic C determined by ^14^C analysis and corrected for Suess and Bomb effects, respectively; Ʊ Total organic carbon (TOC) determined by combustion on an Interscience elemental analyzer EA 1108.

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
