# Peer review of "Organic Matter Preservation in Ancient Soils of Earth and Mars"

_life, 2020, doi:10.3390/life10070113_

Round 1

Reviewer 1 Report

A few comments:

Line 7: “astropedology”. I don’t want to be a stick in the mud, but this term is not really well-established, which then begs the question…is the term needed? Maybe so, but the value should be balanced against the confusion of so many terms for similar science. In fact, in this review article, the term is only used in the first sentence of the abstract and in the keywords (not actually used in the body of the paper at all).

Lines 71-75:
No reference is given for the Soil Science Society 2017 quote.

Line 79: References 25-27 are not ideal for this topic. Reference 25 is another general review article. Reference 26 is good, but it is primarily the evolved gas analyses (not the minerology from ChemMin). Reference 27 is incorrect, I think. The title looks like a 2014 paper, but this says 2018 and the author is different.

Line 138: “both in-situ and orbital investigations observe is the widespread abundance of phyllosilicates (Figures 1 and 2).”  Only Figure 2 shows widespread phyllosilicates, and neither figure addresses abundance of phyllosilicates.

Figure 3: Nowhere in the figure title, figure caption, or text is it stated that these are depth profiles. The reader is expected to understand that, and simply understand that they can be inferred and interpreted to show a loss of carbon over time.

Line 191: Good description of “conversion of smectite to illite”

Line 273 – “However, historical….”  The word historical sometimes implys short timescale (human records, etc). This sentence is referring to much longer times of exposure.

Figure 5 – Too many significant figures in Equation.

Lines 369-373
“The alpha particle x-ray spectrometer (AXPS) onboard Curiosity Rover has the capability to determine FeO / Fe2 O3  of brushed or drilled Martian paleosol profiles, and can analyze crystalline and non-crystalline components of drill fines when combined with x-ray diffraction patterns from the CheMin instrument [29]. Specifically, paleosols rich with Fe/ Mg smectites that show FeO/ Fe2 O3  of greater than one should be considered the highest priority paleosols for biosignature investigation on Mars.”

This is incorrect. APXS only makes elemental measurements, not redox measurements. Also, it does not measure oxygen. Thus, it can reveal the amount of iron, but not how it is bound (as a silicate or oxide). A Mössbauer, as flown on MER rovewrs, can reveal iron redox state, though.

Line 425-427
“A compilation of Cenozoic paleosols with greater than 80 wt. % clay show that organic C is significantly (P<0.0001) correlated with the molecular ratio of CaO + MgO / Al2 O3 (Figure 8).”

This seems like most new aspect of this review paper. More information about this analysis should be provided. From what is given, I simply do not know how to evaluate this. From the figure, it looks like four samples that have higher organic content in Miocene samples than in Eocene samples. With that in mind, P<0.0001 seems stronger than I expect at face value. Also, I am not sure if what I should think it take regarding one of the samples being “Inceptisol.”

Line 446: Wattel-Koekkoek et al. study seems very relevant for this review. More information here would be useful (including what is that data and can it be visualized here).

Line 582-584
The author’s text here follows the hypothesis that there are extensive paleosols in the Farrel Quartzite of Western Australia (Retallack et al., 2016). This hypothesis seems to not (yet?) widely accepted. Reading the comment to the Retallack et al., 2016 and the response, it may be as simple as how extensive soil formation was (or was not). It seems all agree that the FQ started with marine sedimentation. Given that, it seems fairly bold to use this hypothesis in a review artcile without a broader discussion and/or acknowledgment that Sugitani et al. (2016) not don’t agree with the interpretation that the microfossils represent soil organisms.

Lines 593-597 and Figure 11
This seems to be a big part of this paper (not review, but new data), and so some more detail is warranted. First, the author mentions that this data is from gypsic horizons. Is that true of all of the data points? Is the gypsum concentrated in the fossil beds or everywhere? Perhaps add a strat column illustrate where the data comes from with respect to where sulfates are found. Are these data points all different hand samples or different drill holes in one rock or different thin sections or different regions of a few thin sections?
Also, some more detail on the statistical analysis used. What is the equation on the graph?
Also, how were framboids differentiated from other types of pyrite? Maybe, explain Figure 11 D-E so that I can understand it better.
Finally, it looks like the slope and p-value are being driven largely by one organic-rich data point. Are the other data statistically significant without that point. As an aside, I have no doubt that pyrite and organic material are correlated, but it may need to be convincing that the pyrites are framboids for the interpretation to be important for exploring paleosols.

Line 610-611
“(A–C) F118310B = R4336 from A horizon of Jurnpa clay loam, (D-E) F118310E =R4339 from By horizon of Jurnpa clay loam.”
Are these the thin section numbers? Again, other papers on the FQ have not concluded soil formation for this unit, and so this may be a confusing use of terms here. I presume that these rock samples would be referred to as a Precambrian chert by other Precambrian field geologists….is that correct?  Not sure what to suggest here.

Author Response

Please see attachment. Thanks for you insight, and your time.

Reviewer 2 Report

General comments

The manuscript reviews the possibility of paleosoil identification on Mars using observations of paleosoils of the Earth. Although the two planets show several differences, such comparison is important and worth doing, and will became more in the focus as we are progressing toward the Mars Sample Return. Possible paleosoils on Mars might have large importance for astrobiology research. As a result the topic is important for the community of the readers. The structure of the manuscript is good, the language is good, adequate, the illustration fits to the content and provide useful information.

However before publication major revise is necessary. The referee encourages the author to make these changes as the suggested modifications will not require more than 6-8 hours. Main genera suggestions:

  • Although several works have been cited, some more ones would provide a better context for the work. It might be the easiest to put one more paragraph at 79 line (after the soil new definition and before the specific Mars related observations) with mentioning: conditions for weathering and occurrence of liquid water was better in the past than now (Ramirez & Craddock 2018; Bishop et al. 2018), however several models favour an early wet but usually cold surface (https://ui.adsabs.harvard.edu/abs/2015P%26SS..117..401W/abstract) but later decreased the volumetric and temporal occurrence (https://ui.adsabs.harvard.edu/abs/2012AsBio..12..586K/abstract), and during most of the Martian history only microscopic liquid water was probably present (https://ui.adsabs.harvard.edu/abs/2015NatGe...8..357M/abstract, https://ui.adsabs.harvard.edu/abs/2017Icar..282...84P/abstract) with limited possibility of chemical alteration involving, brines and microscopic liquid films (https://ui.adsabs.harvard.edu/abs/2019GeCoA.257..336P/abstract ,https://ui.adsabs.harvard.edu/abs/2019Icar..322..135G/abstract).
  • At the potential weathering processes on Mars, especially related to organics, please provide 1-2 sentences on that UV and ionizing radiation destroy, as well as surface oxidizing agents and their increased importance during the geological history with citing: https://ui.adsabs.harvard.edu/abs/2014IJAsB..13..112D/abstract, https://ui.adsabs.harvard.edu/abs/2007BGeo....4..545D/abstract
  • Many of the readers might not be familiar with the new, modified definition of the “soil” that covers extraterrestrial ones without possible living components. Thus this should be mentioned earlier and even could be indicated already in the abstract (line 7). And despite this new definition, I would suggest to modify their statement there from “recent investigations suggest paleosols may be widespread across the surface of Mars” to “recent investigations suggest paleosol-like features may be widespread across the surface of Mars.”, because these are based on some orbital observations and theoretical argumentation but nt in-situ observations. The community might favour a lighter phrasing as “paleosol-like”.
  • Although phyllosiicate type definition has been published much in various papers for Mars, and they might be real and exact, however it should be noted somewhere that these are the “best candidate types”, which have not been confirmed by ground truth yet. They are probably there, but nobody can be sure exactly the listed ones are the dominant, and not several resemble types are present together. This might be indicated somewhere at the beginning, and will not influence the types listed in the manuscript – but should be noted.
  • The “warm and wet” scenario for the Noachian is poplar, but also need to consider and mention the “wet and cold” Noachian mode (see the specific comments later).
  • It would be favourable to unify the text size in the figures, in some cases the labelling seems to be too large, while in others too small. And try to make diagrams useful and can be interpreted in black and white version too.
  • It would be really useful if a new table would have been implemented that summarizes the Mars relevant observable paleosoil components, for example each line could present one paleosoil component, while different columns might include the characterization of the given component (like chemical formula), method of origin, observability, retention/survival, what does it indicate in a paeosoil.

Specific comments

25 line

“highest priority”

Different researchers might have different order of priority. The topic discussed here is definitely important, but to state as “highest priority” would require to compare it to other high priority aims. Thus the referee suggests to modify to “high priority”, what still means important.

28 line

Please list “astrobiology” among the keywords.

40-41 lines

“Life leaves trace or body fossils in paleosols, and other biosignatures like isotopically light carbon [5] as evidence of their ancient relationship with soils, which may have extended well back into the Archean [6–9].”

Yes, but please consider to mention complex organic material itself.

51-53 lines

Please reformulate this sentence, not clear now.

56 line

“recalcitrant”

Please explain the phrase briefly in bracket, not all readers might be familiar with that.

59 line

“system of thought for targeting”

Not clear, please reformulate.

69 line

“There is mounting evidence that paleosols are preserved”

The referee thinks this should be formulated different, as this statement is based only on orbital measurements, surface in-situ analysis of these candidate paleosols should be made before such statement can be done. Please consider to reformulate to: “There is mounting evidence that paleosols might be present and preserved”

80 line

“investigation of paleosols on Mars”

Consider to reformulate to “investigation of candidate paleosols on Mars”.

89 line

“(By horizon),”

What is “by” for?

91 line

“Sequences of paleosols”

Consider to “Sequences of candidate paleosols”.

Figure 1

(A) and (B) labels are indicated in the caption but are not present in the image. Please indicate them.

115 line

“minerals consistent with those observed”

Do you mean similar mineral types?

119 line

“they resemble lateritic soils”

Please list the specific aspects in which they are similar.

126 line

“warmer and wetter conditions”

Please cosider to mention the possibility of wetter but cold surfce and warmer (internal heted) subsurface”, citing: https://ui.adsabs.harvard.edu/abs/2020Icar..33813567K/abstract, https://ui.adsabs.harvard.edu/abs/2018Icar..300..261P/abstract

130 line

„landing at at Oxia Planum”

Please cite these corresponding works: https://ui.adsabs.harvard.edu/abs/2020SoSyR..54....1I/abstract, https://ui.adsabs.harvard.edu/abs/2016OLEB...46..435K/abstract

133 line, Figure 2

Please explain briefly what is „weathering sequence” with Mars relevance, especially what sequencial observations or indications are there.

158 line

“interpretation paleosols”

Consider to modify to “interpretation of paleosols”.

162 line

“so far proven to be very mild on Mars”

Please reformulate, it is not clear what do you mean.

233-237 line, caption of Figure 4

Please consider to mention Mars relevant aspects of the features in the image.

247 line (around this paragraph)

Please also mention that organics from meteorites also arrived to Mars, especially in the early era and deposited on the surface (cite: https://agu.confex.com/agu/fm15/webprogram/Paper66392.html).

299-305 lines

Please indicate that these very old paleosols might be among the most Mars relevant ones, as they formed by simple orgnaisms.

322-323 lines

This part is quite important, suggest to indicate briefly in the abstract too.

340-341 lines

This part is also quite important, suggest to indicate briefly in the abstract too.

343 line

“mottling”

Please explain.

344 line

“drab-colored”

Please explain.

360 line

“Noachean”

Correct to “Noachian”.

366-368 lines

These are OK, but Fe oxidation state might change in time.

377-379 lines

This part should be more detailed. Can we exclude the reason for smaller early biomass per soil volume unit? And how could this be related to different early organisms than later?

400 line

“extremely resistant to desorption”

Please indicate the difficulty of extraction (separation of clays and organics) is a problem.

405 line

“occlusion”

Please explain the phrase in bracket.

426 line

“clay show that”

Consider to modify to “clay shows that”.

Figure 8

Please use differently shaped symbols, in black and white printing the diagram is difficult to interpret.

474 line

“and water availability lend increased”

Is the phrase “lend” good here?

481 line

“soils lending more surfaces”

Here also is the ”lending” the right term?

486-488 lines

This part is quite important, suggest to indicate briefly in the abstract too.

501 line

“sesquioxides”

Please explain in bracket.

504-505 lines

Do the amorphous phases on Mars have relevance here? (https://ui.adsabs.harvard.edu/abs/2018GeoRL..4512766G/abstract, https://ui.adsabs.harvard.edu/abs/2018Icar..302..285S/abstract)

528-530 lines

Can not understand the relevance and the context of this sentence.

532 line

“imoglite”

Could you explain what it is? Or give formula?

Figure 10

Please use different shaped symbols.

550 line

“Ostwald ripening”

Please explain what is it.

566 line

“which are common products of sulfurization reactions”

Please indicate, were these sulfurization happen during soil formation or later after burial?

621 line

“sulfur content”

It would be useful to mention is the elevated sulfur content favourable or not for organic retention. Please note on Mars more elevated sulfur content can be present in general than on Earth.

640 line

“rich in smectites, amorphous colloids and sulfur”

Please indicate how these components are present / occur in the oldest paleosoils on the Earth.

Author Response

Please see attachment. Thanks for your knowledge, and your time.

Round 2

Reviewer 1 Report

The author has done a good job revising the manuscript from the two reviews.

Reviewer 2 Report

Review of life-810311-R1 manuscript titled

Organic Matter Preservation in Ancient Soils of Earth and Mars

General comments

The revised manuscript became much better, all of the indicated comments have been considered and handled. As a result the referee thinks the manuscript could be accepted. Only few minor issues should be handled:

Please improve the Figure 11 right side: the texts somewhere overlap.

654-657 lines: maybe it should not be bold text

The new supplementary table S1 is very useful, if the related citations would be also indicated here in the table (on the given feature (like Vesicular structure, Gypsum nodules, Calcite nodules) would be even better. And this table might be indicated as a regular figure also (not supplementary).